# Readout of an antiferromagnetic spintronics system by strong exchange coupling of Mn$_2$Au and Permalloy

S. P. Bommanaboyena [1], D. Backes [2], L. S. I. Veiga[2], S. S. Dhesi [2], Y. R. Niu [3], B. Sarpi[3], T. Denneulin [4], A. Kovács [4], T. Mashoff[1], O. Gomonay [1], J. Sinova[1], K. Everschor-Sitte [1], D. Schönke[1], R. M. Reeve [1], M. Kläui [1], H.-J. Elmers [1] & M. Jourdan [1✉]

In antiferromagnetic spintronics, the read-out of the staggered magnetization or Néel vector is the key obstacle to harnessing the ultra-fast dynamics and stability of antiferromagnets for novel devices. Here, we demonstrate strong exchange coupling of Mn$_2$Au, a unique metallic antiferromagnet that exhibits Néel spin-orbit torques, with thin ferromagnetic Permalloy layers. This allows us to benefit from the well-established read-out methods of ferromagnets, while the essential advantages of antiferromagnetic spintronics are only slightly diminished. We show one-to-one imprinting of the antiferromagnetic on the ferromagnetic domain pattern. Conversely, alignment of the Permalloy magnetization reorients the Mn$_2$Au Néel vector, an effect, which can be restricted to large magnetic fields by tuning the ferromagnetic layer thickness. To understand the origin of the strong coupling, we carry out high resolution electron microscopy imaging and we find that our growth yields an interface with a well-defined morphology that leads to the strong exchange coupling.

[1] Institut für Physik, Johannes Gutenberg-Universität, Staudingerweg 7, D-55099 Mainz, Germany. [2] Diamond Light Source, Chilton, Didcot, Oxfordshire OX11 0DE, United Kingdom. [3] MAX IV Laboratory, Fotongatan 8, 22484 Lund, Sweden. [4] Ernst Ruska-Centre for Microscopy and Spectroscopy with Electrons, Forschungszentrum Jülich, D-52425 Jülich, Germany. ✉email: jourdan@uni-mainz.de

A basic concept of antiferromagnetic (AFM) spintronics is to store information by the alignment of the staggered magnetization or Néel vector **N**, typically along one out of two perpendicular easy axes[1–4]. The major benefits of using AFMs as active elements in spintronics are their intrinsically fast THz dynamics[5] and their stability against external magnetic fields, e.g., up to 30 T in the case of $Mn_2Au$[6].

Regarding the manipulation of the Néel vector orientation, the application of short current pulses creating spin-orbit torques (SOT) is a promising approach. These can be created at interfaces with heavy metal layers[7–9] or for metallic compounds in the bulk of the AFM itself[10]. In the latter case, only CuMnAs and $Mn_2Au$ have been identified to combine the required crystallographic and magnetic structure with strong spin-orbit coupling, such that a current along a specific direction can create a bulk Néel spin-orbit torque (NSOT) acting on the Néel vector[10]. Indeed, for both compounds, current pulse-induced magnetoresistance effects of the order of 1% and below were observed and associated with a rotation of the Néel vector[11–15]. Furthermore, the current pulse-induced reorientation of the Néel vector of CuMnAs(001) and $Mn_2Au$(001) was directly demonstrated by magnetic microscopy[16,17]. From these only two available compounds with a bulk NSOT, $Mn_2Au$ stands out concerning potential memory applications due to its metallic conductivity, high Néel temperature (>1000 K)[18], and magnetocrystalline anisotropy, which results in a long term room temperature stability of Néel vector aligned states[6].

Having established current-induced writing, still, the read-out of the Néel vector orientation in a potential device poses a major challenge. The magnitude of the anisotropic magnetoresistance effects (AMR) associated with the reorientation of **N** of metallic antiferromagnets amounts to only 0.1–1%[13,19–22]. The spin-Hall magnetoresistance, most often utilized for insulating AFMs, is even smaller[7–9]. However, most applications require magnetoresistance (MR) effects above 20%[23].

In contrast to the case of AFM spintronics, such MR values are easily obtained in ferromagnetic (FM) spintronics, e.g., based on tunnel magnetoresistance (TMR) of FM/MgO/FM junctions[24,25]. Thus, coupling AFM layers to thin FM films enabling e.g., TMR-based read-out is highly desirable, provided that the fast dynamics of the AFM, as well as its resistance against disturbing magnetic fields, is only moderately diminished. The former requires

sufficiently strong coupling forcing the magnetization to follow the Néel vector, the latter is obtained by using very thin FM layers as we show in this work. Additionally, the sensitivity to external fields could be further reduced by replacing the single FM layer by a synthetic antiferromagnetically coupled FM bilayer with zero net magnetization[26].

Here, we demonstrate a strong exchange coupling of 40-nm-thick $Mn_2Au$(001) epitaxial thin films with very thin (down to 2 nm) Permalloy (Py) layers. The coupling of the magnetization vector $\mathbf{M}_F$ of the FM and of **N** results in the perfect imprinting of the AFM domain pattern of the $Mn_2Au$ film on the FM domain pattern of the soft Py layer. Similar imprinting was only reported for insulating AFM/ metallic FM bilayers such as 1.2 nm of Co on 40 nm of $LaFeO_3$[27] or Fe(5 nm) on NiO (15 nm)[28]. However, in these cases, it was associated with a coercive field of only ≃100 Oe. In contrast, we show that 5000 Oe are required to reverse the magnetization of 2 nm of Py on 40 nm of $Mn_2Au$(001). This, at room temperature, represents a coercive field corresponding to a field stability range, which is more than an order of magnitude larger than that in related CuMnAs/Fe(2 nm) bilayers, where a long term stable remanent magnetization was reported measuring at 200 K[29]. We show that in $Mn_2Au$(001)/Py even at the coercive field of 5000 Oe $\mathbf{M}_F$ and **N** do not decouple, but rotate together driven by the Zeeman energy of the FM layer. Thus, $Mn_2Au$(001)/Py represents an excellent system for read-out in AFM spintronics. Furthermore, we identify the morphological origin of the exceptionally strong magnetic coupling between these AFM and FM layers.

## Results

**$Mn_2Au$/Py samples.** $Mn_2Au$ has a tetragonal crystal structure and orders antiferromagnetically well above 1000 K[18]. It shows collinear AFM order consisting of antiparallel stacked planes with FM order, as indicated in Fig. 1c. Within the easy (001)-plane, there are four equivalent easy <110> -directions, which results for our as-grown $Mn_2Au$(001) thin films in the formation of AFM domains with a typical size of 1 μm corresponding to all four associated orientations of **N**[6].

Here, we investigate $Mn_2Au$(001)(40 nm)/$Ni_{80}Fe_{20}$ (Py) (2 to 10 nm) bilayers, which are grown on $Al_2O_3$(r-plane) substrates with a Ta(001)-buffer layer and a capping layer of 2 nm of $SiN_x$. Scanning transmission electron microscopy with high-angle

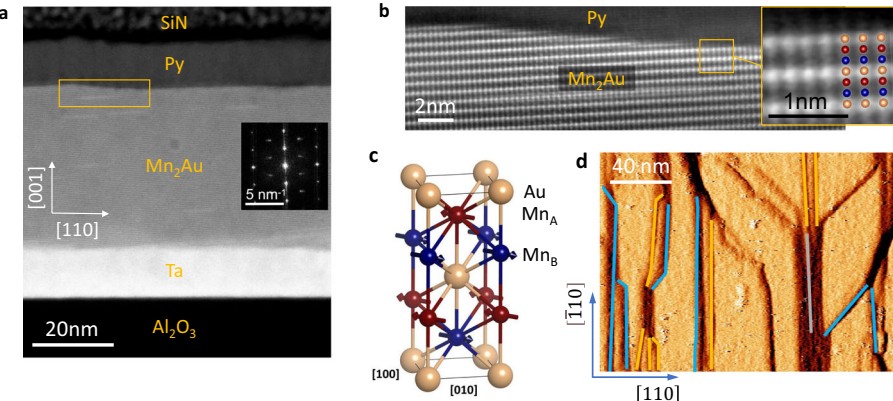

**Fig. 1 Structure of the samples. a** Cross section HAADF STEM image of the entire stack viewed along the [1̄10]-direction of $Mn_2Au$(001). The inset shows a local Fourier transform of the $Mn_2Au$ region. **b** Magnified image of the $Mn_2Au$/Py interface (region indicated by a rectangle in **a**), where Au atom columns have bright contrast. The inset shows a higher magnification image overlaid with a model of the crystal structure. **c** Crystal structure of $Mn_2Au$ with the magnetic moments pointing along the easy [1̄10]-direction. **d** STM image of a pristine $Mn_2Au$(001) thin film surface with steps corresponding to half unit cells (0.42 nm) indicated by the yellow lines, one unit cell (0.85 nm) indicated by the blue lines, and three unit cells (2.55 nm) indicated by the gray line.

annular dark-field imaging (STEM-HAADF) of the complete multilayer is shown in Fig. 1a. The Fourier transform of the epitaxial $Mn_2Au(001)$ thin film region (inset in Fig. 1a) shows regularly-spaced Bragg peaks, which confirm that the layer is monocrystalline. Most importantly, Fig. 1b shows a magnification of the interface between $Mn_2Au(001)$ and Py, where we find that our growth leads to a defined Au termination of the AFM thin film. This is supported by UHV STM-images of a pristine $Mn_2Au$ surface, which show atomically flat terraces with steps corresponding to the half or the full length of the c-axis (Fig. 1d). Such steps are consistent with the well-defined Au termination. This, as we demonstrate later, implies that the same AFM sub-lattice couples to the FM at the interface, which generates a very strong coupling and leads to a one-to-one correspondence of the AFM and FM domain patterns.

**Hysteresis loops**. We quantified the coupling between the $Mn_2Au(001)$ and Py layers by measuring magnetic hysteresis loops. Single Py films are in general magnetically soft with coercive fields of the order of a few Oe, which, as we will show below, drastically increases if they are coupled to AFM $Mn_2Au$ films.

Figure 2 shows hysteresis loops of a $Mn_2Au(40\,nm)/Py(4\,nm)$ bilayer measured in a superconducting quantum interference device (SQUID) with the easy [110] axis of the $Mn_2Au(001)$ thin film aligned parallel to the magnetic field direction.

After the as-grown sample consisting of an AFM multidomain state was placed in the SQUID, subsequent hysteresis loops with increasing maximum fields from 300 to 1000 Oe were measured as shown in the left inset of Fig. 2. In this field range, we obtained smooth hysteresis loops with almost zero remanent magnetization. This is consistent with a strong coupling of the Py magnetization to the AFM domain configuration of $Mn_2Au$, which either remains unaffected by the magnetic field or restores the original domain configuration when the field is zero again.

However, once the magnetic field exceeds a threshold value, square-shaped easy axis loops with a coercive field $H_c \simeq 1600$ Oe were obtained, as shown in Fig. 2. This behavior can be readily explained by assuming a strong exchange coupling of $\mathbf{M}_F$ and $\mathbf{N}$, which results in the Zeeman energy of the FM layer driving a reorientation of $\mathbf{N}$. Measuring hysteresis loops of $Mn_2Au(40\,nm)/$ Py bilayers with various Py thicknesses varying from 2 to 10 nm, we observed linear scaling of the coercive field $H_c$ with the inverse saturation magnetic moment $1/m_F$, as shown in the right inset of Fig. 2. This is consistent with the assumption that the Zeeman energy $\mathbf{M}_F \cdot \mathbf{H}$ of the Py layer sets the remanent reorientation of both $\mathbf{M}_F$ and $\mathbf{N}$.

To check if this holds, we next probe the orientation of the magnetization and the Néel vector:

**Probing the Néel vector orientation**. We investigated the orientation of $\mathbf{N}$ and $\mathbf{M}_F$ of a $Mn_2Au(40\,nm)/Py(4\,nm)/$ $SiN_x(2\,nm)$ sample by x-ray absorption spectroscopy (XAS), in the surface-sensitive total electron yield (TEY) as well as in the bulk sensitive substrate fluorescence yield (FY) mode. The magnetic contrast was obtained based on the x-ray magnetic linear and circular dichroism (XMLD/XMCD) effects, as described in the Methods section (the experimental geometry is shown in Fig. 3a).

In the as-grown state of the sample, due to the relatively large diameter of the x-ray beam ($\simeq$500 μm), we average over many AFM/FM domains canceling both the XMLD and XMCD to zero. We then apply a magnetic field of 1 T (i.e., well below the spin-flop field of $\simeq$30 T[6,20]) parallel to the x-axis, i.e., to the easy [110]-direction of $Mn_2Au(001)$ and subsequently reduce the field to zero again. In agreement with the hysteresis loops discussed above, this results in a significant Ni-$L_{2,3}$ edge XMCD signal (measured at a tilt angle of $\Theta = 60°$) showing a sizable remanent in-plane magnetization of Py (Fig. 3b). Additionally, we demonstrate that the Néel vector of the AFM becomes aligned as well, both at the interface and in the bulk parallel to $\mathbf{M}_F$ of the FM. The corresponding XMLD spectra (measured at $\Theta = 0°$) are shown in Fig. 3, obtained in the surface-sensitive TEY mode (Fig. 3c) as well as in the bulk sensitive substrate FY mode (Fig. 3d). From the characteristic sign change of the XMLD signal, we can conclude that $\mathbf{N}$ becomes aligned parallel to $\mathbf{M}_F$ as deduced by comparison with previous results[30]. After applying

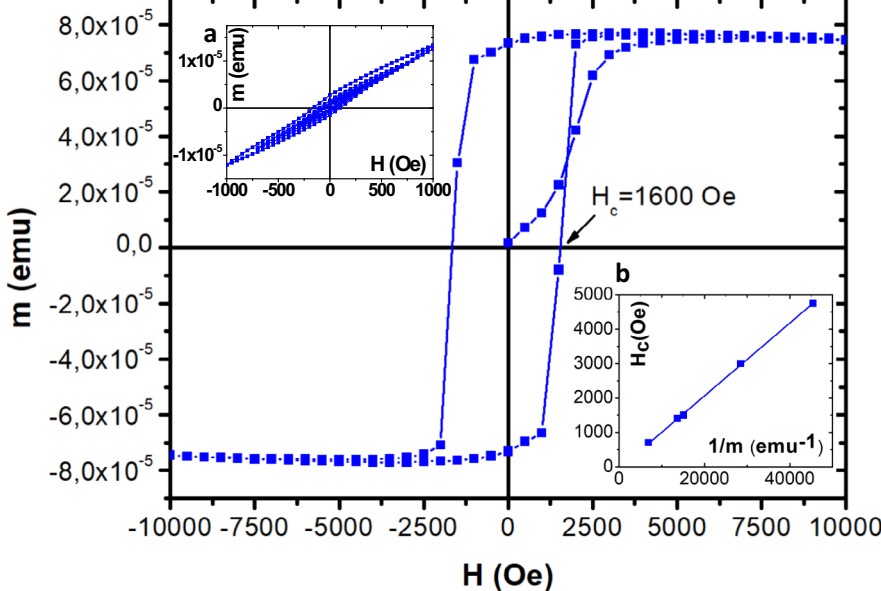

**Fig. 2 Hysteresis loops.** Inset **a** shows subsequent hysteresis loops, i.e., measurements of the magnetization **m** of a $Mn_2Au(001)(40\,nm)/Py(4\,nm)$ sample taken with increasing applied magnetic fields **H** up to $10^3$ Oe (aligned parallel to the easy [110] direction of $Mn_2Au$). The main panel shows the subsequent loop up to $10^4$ Oe (also aligned parallel to [110]). Inset **b** shows the coercive field **H$_c$** vs. the inverse saturation magnetization (with linear fit) obtained from $Mn_2Au(40\,nm)/Py$ bilayers with the Py thickness varying from 2 to 10 nm.

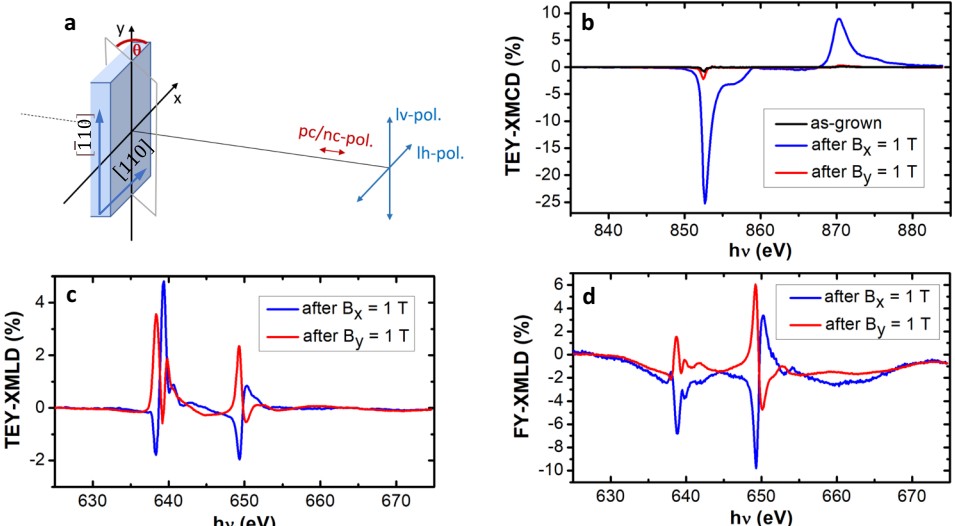

**Fig. 3 XMCD/XMLD of Mn$_2$Au(40 nm)/Py(4 nm)/SiN$_x$(2 nm).** Signal averaged over the x-ray illumination spot of $\simeq$360 μm$^2$ × 530 μm$^2$. **a** Experimental geometry showing the x-ray polarization directions and orientation of the Mn$_2$Au(001) epitaxial thin film. For measuring the XMCD with in-plane sensitivity, the sample is rotated by $\Theta = 60°$ around the y-axis. **b** TEY-XMCD at the Ni-L$_{2,3}$-edge obtained with $\Theta = 60°$. **c** TEY-XMLD at the Mn-L$_{2,3}$-edge obtained from the difference between XAS recorded with linear horizontal (lh) and XAS with linear vertical (lv) x-ray polarization, $\Theta = 0°$. **d** Bulk sensitive substrate FY-XMLD at the Mn-L$_{2,3}$-edge, $\Theta = 0°$.

the magnetic field parallel to the [$\bar{1}$10]-direction of Mn$_2$Au(001) and subsequently reducing the field to zero, the XMCD ($\Theta = 60°$) becomes zero since the remanent magnetization is oriented perpendicular to the direction of photon incidence. Consistently, the XMLD signal ($\Theta = 0°$) is inverted both at the interface and in the bulk as expected for a 90° rotated Néel vector.

With this strong parallel coupling of **N** and **M**$_F$, we expect the AFM domain pattern of Mn$_2$Au, as formed during thin film growth, to be imprinted into the FM domain pattern of the Py layer, which we verify by XMLD-photoelectron emission microscopy (PEEM).

**Imaging the AFM/FM domain pattern.** XMLD-photoelectron emission microscopy (PEEM) of the FM domain pattern of the Py layer and simultaneously of the AFM domain pattern of the Mn$_2$Au(001) layer of a Mn$_2$Au(40 nm)/Py(4 nm)/SiN$_x$(2 nm) sample was performed with a perpendicular incidence of the photon beam, as described in the Methods section.

We first discuss samples in the as-grown state, which were not exposed to a magnetic field. As shown in Fig. 4, panels a (AFM domains) and b (FM domains), the AFM domain pattern of Mn$_2$Au(001) is perfectly imprinted on the FM domain pattern of the Py layer. Note that we have chosen also the FM XMLD contrast mechanism. As a result, also for the FM domains, only the axis along which **M**$_F$ is aligned produces a brightness contrast, not its direction, allowing for a direct comparison.

The size of the AFM/FM domains of $\simeq$1 μm, is at least one order of magnitude larger than the typical distance between the steps at the surface shown in Fig. 1. Due to the well-defined Au termination of the Mn$_2$Au(001) layer as discussed above, crossing a morphological step at the interface does not change the AFM sub-lattice to which the **M**$_F$ of Py couples. This enables a strong planar exchange coupling at the interface, as indicated in Fig. 5, which we address in more detail in the Discussion section.

Due to the associated one-to-one correspondence of the AFM and FM domain pattern, we can indirectly obtain microscopic images of the Mn$_2$Au AFM domains using scanning electron microscopy with polarization analysis (SEMPA)[31] (Fig. 4c–e).

Horizontal (Fig. 4c) and vertical (Fig. 4d) in-plane components of **M**$_F$ of Py are measured separately, revealing the direction of **M**$_F$ (Fig. 4e). Furthermore, SEMPA shows the same characteristics of the domain structure as the XMLD-PEEM images discussed above, but with the experimental advantage of the enhanced availability of a lab-based technique.

By SEMPA, we also verified that Mn$_2$Au(40 nm)/Py(4 nm)/SiN$_x$(2 nm) samples, which were previously exposed to magnetic fields above $H_c$ (see Fig. 2), are homogeneously magnetized.

## Discussion

The initial AFM domain pattern of the 40 nm Mn$_2$Au(001) thin films forms during growth driven by an intrinsic mechanism such as the minimization of elastic energy[32] and couples to the Py magnetization during the room temperature deposition of this thin layer. No exchange bias is present or obtainable in the bilayers investigated here, because of the high Néel temperature >1000 K of Mn$_2$Au[18], which prevents the creation of exchange bias by standard field cooling procedures[33].

Nevertheless, for a discussion of the magnetic coupling mechanism of the Mn$_2$Au/Py bilayers, the physics of exchange bias systems[34] is a good starting point. Such systems can be used as references for the magnitude of the coupling of the Py magnetization **M**$_F$ to the Néel vector **N** of Mn$_2$Au(001): The coercive fields presented above are at least one order of magnitude larger than those of typical exchange bias systems with a comparable Py-layer thickness[35–38]. Regarding other work on Mn$_2$Au/FM bilayers, coercive fields of the order of only 100 Oe or less were reported, which could be related to different growth directions of these Mn$_2$Au thin film[39,40] as well as to the selection of Fe as the FM layer[33,39], or do a different morphology.

An explanation for the substantially different coercive fields is obtained by considering the morphology of the interface between the AFM and FM layers. Whereas in the general framework of exchange bias, typically interface irregularities of the AFM such as roughness or random alloy effects associated with uncompensated spins are discussed as pinning centers for FM domain walls[41–45], a planar exchange interaction at the AFM-FM interface is able to couple the Néel and magnetization vectors of the

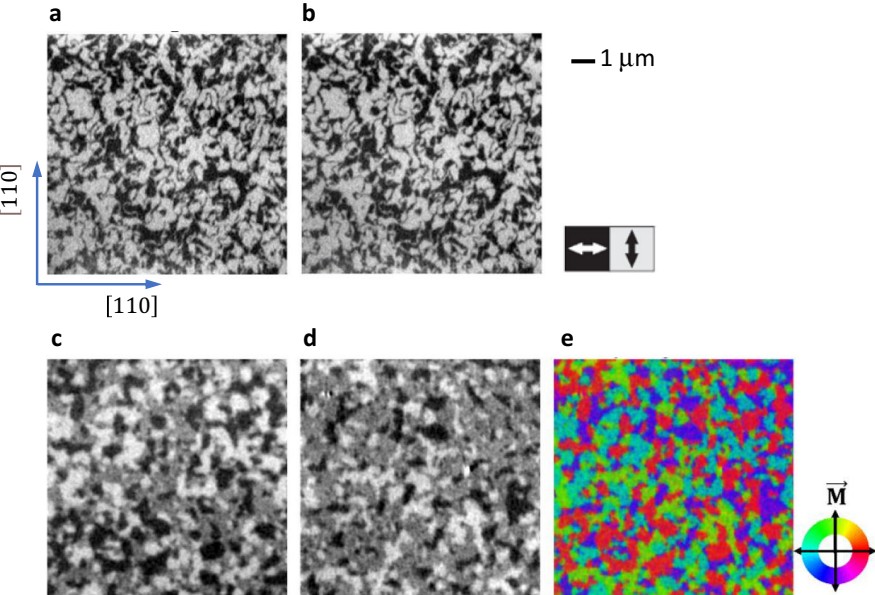

**Fig. 4 Magnetic microscopy of as-grown Mn₂Au(40 nm)/Py(4 nm)/SiNₓ(2 nm).** The horizontal and vertical edges correspond to the four easy <110> directions of the epitaxial Mn₂Au(001) thin film. The field of view is 10 μm² × 10 μm². **a** XMLD-PEEM image of the Mn₂Au AFM domains in the as-grown state. **b** Corresponding XMLD-PEEM image of the Py FM domains. Both in **a** and **b** dark corresponds to **N** and **M**$_F$ right-left and bright to up-down. **c** SEMPA image showing the x-component of the FM contrast. **d** SEMPA image showing the y-component of the FM contrast. **e** is generated from **d** and **e** showing the orientation of **M**$_F$ by a color code, which indicates that **M**$_F$ of the Py domains is pointing parallel to all four easy <110> directions of the Mn₂Au(001) thin film.

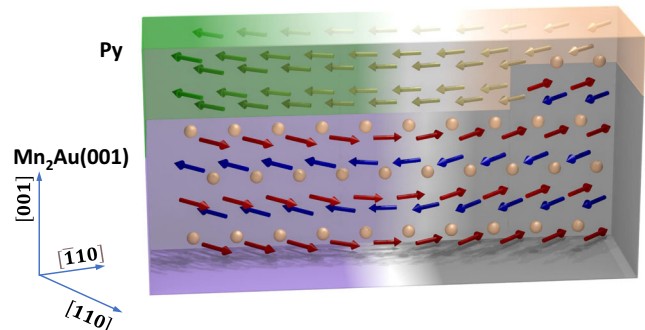

**Fig. 5 Magnetic coupling in Mn₂Au(001)/Py bilayers.** This schematic cut through the bilayer shows a side view of two 90° domains and visualizes the morphological origin of the imprinting of the AFM domain pattern into the FM. Due to the specific interface morphology with a well-defined Au termination, **M**$_F$ is always aligned parallel with **N** and couples with the same relative orientation, despite the existence of atomic steps.

respective layers much more strongly[46]. This necessarily requires ferromagnetically ordered atomic layers of one defined AFM sub-lattice forming the surface of the AFM. Nevertheless, just the appropriate type of magnetic ordering is not sufficient, as depending on the interface morphology atomic steps can result in an alternating sign of the exchange coupling if different sub-lattices are present at the AFM/FM interface.

However, we find that our growth of the Mn₂Au(001)/Py bilayers leads to a well-defined Au termination of the surface of the Mn₂Au(001) thin films (Fig. 1). This means that everywhere at the surface the same AFM sub-lattice couples to the FM resulting in the same sign of exchange coupling over each AFM domain as shown schematically in Fig. 5. This leads to a maximum coupling strength and thus enables the perfect imprinting of the AFM domain pattern of the Mn₂Au(001) thin film on the

FM domain structure of the Py layer, which by itself would be magnetically highly isotropic and soft.

The coupling of the Py magnetization **M**$_F$ to the Néel vector **N** is so strong that the layers are not even decoupled at the large coercive field but jointly rotated. We describe this reorientation under the action of the magnetic field using a macrospin model. Within this model, we describe the density of the magnetic energy per unit area of the Mn₂Au/Py bilayer by

$$w(\mathbf{M}_F, \mathbf{N}) =$$
$$-\frac{H_{an}t_{AF}}{M_s^3}\left(N_x^4 + N_y^4\right) - J_{coup}\xi\mathbf{M}_F \cdot \mathbf{N} - t_F\mathbf{M}_F \cdot \mathbf{H}, \quad (1)$$

where the first term contains the magnetocrystalline anisotropy energy of Mn₂Au ($H_{an} > 0$ is the in-plane anisotropy field, $M_s = |\mathbf{N}|$), and the second term describes the exchange coupling of the Néel and Py magnetization vectors. $t_{AF}$ and $t_F$ are the thicknesses of the antiferromagnetic and ferromagnetic layers, $J_{coup} > 0$, and $\xi$ describe the strength and the characteristic length of the exchange coupling between the AFM and FM layers. The third term represents the Zeeman energy of the Py layer in a magnetic field **H**. The coordinate axes $x$ and $y$ are aligned along the easy magnetic axes of Mn₂Au.

The equilibrium configurations are obtained by minimizing (1) with respect to the magnetic vectors **M**$_F$ and **N**. In zero field **M**$_F$↑↑**N** are both parallel to $\hat{x}$ or $\hat{y}$. A magnetic field **H** ↑↑ $\hat{x}$ splits the degeneracy between the states with **M**$_F$↑↑**H**, **M**$_F$↑↓**H**, and **M**$_F$⊥**H** and creates a ponderomotive force ∝ $BM_F$, which acts on the domain walls[47]. This force shifts the domain walls thus reducing the fraction of the energetically unfavorable states (domains). This process shows up in the smooth growth of the magnetization as shown in Fig. 2a. As long as the value of the applied field is not sufficient to remove all domain walls from the sample, the process is reversible, corresponding to the minor hysteresis loops.

However, above a threshold field value, the sample develops a single domain state and further cycling of the magnetic field

induces switching between the states $\mathbf{M}_F\uparrow\uparrow\mathbf{H}$ and $\mathbf{M}_F\uparrow\downarrow\mathbf{H}$. From the stability conditions for these states we obtain the coercive field

$$H_{\text{coer}} = \frac{4H_{\text{an}}J_{\text{coup}}M_s\xi t_{\text{AF}}}{4H_{\text{an}}t_{\text{AF}} + J_{\text{coup}}M_F\xi}\frac{1}{t_{\text{F}}}. \qquad (2)$$

In the limit of strong exchange coupling, $J_{\text{coup}}M_F\xi \gg H_{\text{an}}t_{\text{AF}}$, the coercive field $H_{\text{coer}} \to 4H_{\text{an}}M_st_{\text{AF}}/(M_Ft_{\text{F}})$ is associated with the magnetocrystalline anisotropy of the $Mn_2Au$ layer and shows a linear dependence on the inverse magnetization $1/m_F = 1/(M_Ft_{\text{F}})$ as observed experimentally (see Fig. 2b).

Inserting the experimentally determined magnetization of the Py layers of $M_F = 1.8\mu_B$ per $Ni_{80}Fe_{20}$-atom and taking $M_S = 4\mu_B$ per Mn-atom from[18], we obtain $H_{an} \simeq 40$ Oe, which corresponds to an anisotropy energy $K_4 = 2 \cdot 4\mu_B \cdot 40 \cdot 10^{-4}T = 1.8\,\mu\text{eV}$ per formula unit (f.u.). This is in good agreement with our previous estimation of $K_4 > 1\,\mu\text{eV/f.u.}$[6], thereby supporting the validity of our model assumptions.

$H_{\text{coer}}$ should also increase with the thickness $t_{\text{AF}}$ of the $Mn_2Au$ layer, which we indeed observed experimentally (e.g., $H_{\text{coer}} = 2000$ Oe for $Mn_2Au(60\,\text{nm})/Py(4\,\text{nm})$). However, this increase is not linear, presumably due to a thickness dependence of the crystallographic order.

In summary, we demonstrated an extremely strong exchange coupling between the Néel vector $\mathbf{N}$ of the metallic antiferromagnet $Mn_2Au$ and the magnetization $\mathbf{M}_F$ of ferromagnetic Py thin films. This results in the perfect imprinting of the AFM domain patterns of 40 nm $Mn_2Au(001)$ thin films on the FM domain pattern of thin (2–10 nm) Py layers, while maintaining high stability in external magnetic fields. The strong coupling results from the particular morphology that we obtain from our growth at the $Mn_2Au/Py$ interface, identified by TEM and STM, where within every domain always the same AFM sub-lattice couples to the FM thus maximizing the exchange coupling. This strong coupling of $\mathbf{N}$ and $\mathbf{M}_F$ enables the electric detection of the Néel vector orientation via standard techniques used for ferromagnetic thin films, thereby providing a solution for the challenging read-out in antiferromagnetic spintronics.

## Methods
All layers of the $Al_2O_3$(r-plane)/Ta(001)(13 nm)/$Mn_2Au(001)$(40 nm)/$Ni_{80}Fe_{20}$ (Py)(2 to 10 nm)/$SiN_x$(2 nm) samples were prepared by rf sputtering. The epitaxial Ta(001) buffer layers were sputtered with a substrate temperature of 700 °C, the epitaxial $Mn_2Au(001)$ thin films were deposited at 500 °C followed by an annealing procedure at 700 °C as described in refs. [48,49]. All other layers are polycrystalline and were deposited at room temperature. $SiN_x$(2 nm) sputtered from a $Si_3N_4$ target in Ar provides a very effective capping layer, which prevents sample oxidation while being highly transparent for photoemitted electrons[50].

The magnetic hysteresis loops of the samples were measured in a Quantum Design MPMS SQUID-magnetometer.

The antiferromagnetic order in $Mn_2Au$ causes an x-ray magnetic linear dichroism (XMLD), i.e., an asymmetry in the absorption of linear polarized x-rays at the $Mn-L_{2,3}$-edge for the polarization direction parallel and perpendicular to $\mathbf{N}$[30]. It is sensitive to the components of $\mathbf{N}$, which are parallel versus perpendicular to the electric field vector of the x-ray beam, but it does not change sign upon reversal of $\mathbf{N}$. Thus we were able to investigate the orientation of $\mathbf{N}$ of a $Mn_2Au(40\,\text{nm})/Py(4\,\text{nm})/SiN_x(2\,\text{nm})$ sample by x-ray absorption spectroscopy (XAS) at the I06 branch line at Diamond Light Source, which is equipped with a superconducting vector magnet. In parallel to probing the Néel vector orientation by XMLD, we were also able to probe the orientation of the magnetization of the Py layer using the x-ray circular dichroism (XMCD) at the $Ni-L_{2,3}$ edge. The XMCD is an asymmetry in the absorption of circular polarized x-rays determined by the component of $\mathbf{M}_F$ parallel to the direction of the incoming x-ray beam[51]. It does change sign upon reversal of $\mathbf{M_F}$. In the surface-sensitive total electron yield (TEY) mode, the sample current induced by the photoemitted electrons with an escape depth of 2–3 nm is probed[52]. In the bulk sensitive substrate fluorescence yield (FY) mode, the measurement signal originates from the x-rays transmitted through the magnetic layer stack generating fluorescence in the MgO substrates, which is probed by a photo diode[53]. The experimental geometry is shown in Fig. 3a.

XMLD-photoelectron emission microscopy (PEEM) was performed with a perpendicular incidence of the photon beam at the MAXPEEM beamline of MAX IV, Lund. The FM XMLD contrast was obtained at the $L_3$ edge of Fe[54], the AFM XMLD contrast at the $L_3$ edge of Mn[30]. The SPELEEM microscope (Elmitec GmbH) of the MAXPEEM beamline provides, due to its perpendicular photon incidence on the sample surface, the highest possible AFM contrast for the in-plane orientation of the Néel vector of $Mn_2Au$.

A SEMPA[31] differs from an ordinary SEM in that it is additionally equipped with a spin detector to analyse the net spin of outgoing secondary electrons from ferromagnetic materials. This enables a spatial mapping of the spin polarization and by extension of the domain structure with a spatial resolution of less than 30 nm. As a ferromagnet possesses an inherent spin asymmetry in its density of states, the outgoing population of secondary electrons (which are ejected from a broad binding energy range) is spin-polarized. These secondary electrons undergo spin-dependent scattering at a W(100) single crystal via spin-polarized low-energy electron diffraction (SPLEED)[55]. Thus, up and down spin electrons are preferentially deflected towards separate electron counters and the difference of the count rates reflects the local magnetization direction.

## Data availability
The magnetic hysteresis (Fig. 2) and XMCD/XMLD (Fig. 3) data generated in this study have been deposited in the Zenodo database under accession code https://doi.org/10.5281/zenodo.5588122.

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

## Acknowledgements

Funded by the Deutsche Forschungsgemeinschaft (DFG, German Research Foundation) - TRR 173 - 268565370 (projects A01, A05, and B12), by the Horizon 2020 Framework Program of the European Commission under FET-Open Grant No. 863155 (s-Nebula), and by the European Research Council (ERC) under the Horizon 2020 research and innovation program Grant No. 856538 (3D MAGIC). K.E.-S. acknowledges funding from the DFG project No. 320163632. We acknowledge MAX IV Laboratory for time on Beamline MAXPEEM under Proposal 20200160. Research conducted at MAX IV, a Swedish national user facility, is supported by the Swedish Research council under contract 2018-07152, the Swedish Governmental Agency for Innovation Systems under contract 2018-04969, and Formas under contract 2019-02496. We acknowledge Diamond Light Source for time on I06-1 under proposal MM29305. We would like to thank S. Jenkins for preparing Fig. 5.

## Author contributions

S.P.B. and M.J. primarily wrote the paper with contributions from O.G. and M.K.; S.P.B prepared the samples and performed the SQUID as well as the SEMPA investigations; D.B., S.S.D., and L.S.I.V. performed and evaluated the XMCD-/XMLD-experiments; Y.R.N. and B.S. obtained the PEEM images; T.D. and A.K. contributed the STEM investigation; T.M. obtained the STM image; D.S. and R.M.R. supported the SEMPA investigations; H.-J.E., K.E.-S., and J.S. contributed to the discussion of the results; M.J. coordinated the project.

## Funding

## Competing interests

The authors declare no competing interests.
