## [Peer Review File · Nature Communications]

Reviewers' Comments:

Reviewer #1:

Remarks to the Author:

The authors use both magnetic hysteresis loops and imaging methods (XMLD-PEEM and SEMPA) to characterize the strong coupling between antiferromagnet Mn₂Au and ferromagnet Py. They ascribe the origin to the Au termination-induced the same sublattice in all over of the interface, which was observed by STEM and STM. This work is quite interesting, and the high quality of the samples should be quite critical for the observation. We recommend the acceptance of this manuscript in Nature communications after adequate revisions. Here, we provide several suggestions and comments of this manuscript.

1. According to the equation (2), the coercive field is linear to the thickness of Mn₂Au. Thus, Mn₂Au thickness-dependent magnetic hysteresis loops are helpful to support the analysis. Please show hysteresis loops of one or two more samples with different Mn₂Au thickness.
2. Hysteresis loops with positive/negative field cooling are also helpful to understand the strong coupling effect. Please show hysteresis loops of Mn₂Au/Py under positive and negative field cooling, whether exchange bias would appear and show opposite sign under different directional field cooling?
3. XMCD is widely used to characterize ferromagnets, as shown in figure 3b. Why XMCD signal after B_y is nearly zero rather than opposite to the signal after B_x, is it related to the measurement configuration ($\Theta = 60$ degree rather than $\Theta = 0$)? Why $\Theta = 60$ degree is chosen for XMCD while $\Theta = 0^\circ$ is chosen for XMLD? If the XMCD signal after B_y is theoretically near zero, the average XMCD signal of as-grown sample should not be zero. Please explain the result of figure 3b.
4. In the line 252, authors describe that "By SEMPA, we also verified that Mn₂Au(40 nm)/Py(4 nm)/ SiNx(2 nm) samples, which were previously exposed to magnetic fields above H_c (see Fig. 2), are homogeneously magnetized". While the experimental evidence is absent. Maybe some experimental data or more discussions are necessary.
5. It seems that the coercive field of Mn₂Au (100)/Py in this work is much larger than Mn₂Au (100)/Fe (Ref. 31, Adv. Mater. 24, 6374 (2012)), as well as Mn₂Au (100)/Py and Mn₂Au (100)/[Co/Pd] (Nat. Mater. 20, 800 (2021)) at room temperature. Whether the difference is from the quality of the Mn₂Au films, especially the FM/AFM interface? Please add some explanation in the revised manuscript.

Then, we also present several minor comments.

1. The title "a antiferromagnetic" should be replaced by "an antiferromagnetic", similarly, revising the spelling errors.
2. In the line 238, author describe that "This enables a strong planar exchange coupling at the interface, as indicated in Fig. 4c." Maybe the "Fig. 4c" should be replaced by "Fig.4", that the coupling is obtained from all panels in figure 4.
3. Whether SEMPA imaging of figure 4c-e is from the as-grown state or after magnetic field? Please add description in the figure caption.
4. According to this manuscript, XMLD can be used to characterize ferromagnet, which is not common. What's the difference between XMLD and XMCD for the characterization of ferromagnetic moments? Why XMCD is shown in figure 3b, while XMLD is chosen in figure 4b?

Reviewer #2:

Remarks to the Author:

The paper by Bommanaboyena et al. reports on the study of exchange coupling in a bilayer system of antiferromagnetic (AFM) Mn₂Au and ferromagnetic (FM) Permalloy metals. In the study the authors observed a large coercive field for Py films (of the order of a few thousands of Oe) when coupled to the AFM. This strong coupling is further supported by a combination of x-ray magnetic linear dichroism (XMLD) and circular dichroism (XMCD) effects which showed a good correspondence of the AFM domain patterns (in Mn₂Au) to the FM domain pattern (in Py). The authors argued that the large coercive field of Py is due to the magnetocrystalline anisotropy of the Mn₂Au layer and observed it to depend linearly on the inverse magnetization of Py. This is a solid piece of work, which will be of interest to researchers in magnetism, especially those working with exchange bias. It certainly should be published in some form. However I disagree

with the main motivation point in the present version of the manuscript; e.g., in the abstract, the authors want to benefit from the well-established read-out methods of ferromagnets, while the essential advantages of antiferromagnetic spintronics are retained. In my view, the main advantage is the fast dynamics of the AFM and adding FM parts back would remove this advantage; or one has to prove experimentally that an AFM coupled to FM would retain its fast dynamics.

Reviewer #3:

Remarks to the Author:

The author successfully developed Mn₂Au/NiFe heterostructure with large exchange bias. The results are very useful to make this class of materials into real applications. In addition, they also revealed the mechanism behind this large exchange coupling and also demonstrated the large exchange coupling leads to the one-to-one imprinting of the antiferromagnetic on the ferromagnetic domain pattern, as well as the alignment of the NiFe magnetization reorients the Mn₂Au Neel Vector. This experimental evidence is very strong, and analyses are sound. I recommend the publication. One minor suggest is to indicate that Fig.3c and Fig.3d are for Mn L edge.

Point-by-point response

We would like to thank the referees for their helpful reports. Below, we supply a point-by-point response regarding all critical comments given by the referees (printed in italics).

All modifications in phrases cited from our manuscript are printed in blue letters, here as well as in the revised version of our manuscript.

Reviewer #1

1. According to the equation (2), the coercive field is linear to the thickness of Mn₂Au. Thus, Mn₂Au thickness-dependent magnetic hysteresis loops are helpful to support the analysis. Please show hysteresis loops of one or two more samples with different Mn₂Au thickness.

We observe an increase of the coercitive field with the Mn₂Au layer thickness as shown in the graph on the right for Mn₂Au layers with d=25 nm, 40nm, and 60nm (Py is always 4 nm). We also observe that the XRD rocking curve is wider for thinner samples. Furthermore, a weak impurity phase is observable in XRD [46], which relative to the Mn₂Au(001) XRD-Peak gets stronger for thinner films. This could explain why samples with thinner Mn₂Au layer have relatively large coercitive fields corresponding to larger effective anisotropy constants.

We added the following statement to the end of the Discussion section:

“ H_{coer} should also increase with the thickness of the Mn₂Au layer, which we indeed observe experimentally (e. g. $H_{\text{coer}} = 2000$ Oe for Mn₂Au(60 nm)/Py(4 nm)). However, this increase is not linear, presumably due to a thickness dependence of the crystallographic order.”

2. Hysteresis loops with positive/negative field cooling are also helpful to understand the strong coupling effect. Please show hysteresis loops of Mn₂Au/Py under positive and negative field cooling, whether exchange bias would appear and show opposite sign under different directional field cooling?

It is not possible to obtain an exchange bias of Mn₂Au/Py bilayers by standard field cooling procedures, as these require to cool the sample through the antiferromagnetic ordering temperature, which in the case of Mn₂Au is well above the decomposition/ melting temperature of the compound. To clarify this, we now write at the beginning of the DISCUSSION section:

“No exchange bias is present or obtainable in the bilayers investigated here, because of the high Néel temperature > 1000 K of Mn₂Au [18], which prevents the creation of exchange bias by standard field cooling procedures [31]”.

3. XMCD is widely used to characterize ferromagnets, as shown in figure 3b. Why XMCD signal after B_y is nearly zero rather than opposite to the signal after B_x , is it related to the measurement configuration ($\Theta = 60$ degree rather than $\Theta = 0$)? Why $\Theta = 60$ degree is chosen for XMCD while $\Theta = 0^\circ$ is chosen for XMLD? If the XMCD signal after B_y is theoretically near zero, the average XMCD signal of as-grown sample should not be zero. Please explain the result of figure 3b.

XMCD is sensitive to the component of the FM magnetization parallel to the momentum direction of the incoming x-rays. The XMCD signal is reversed, if the magnetization direction is reversed

(antiparallel instead of parallel to the direction of the x-ray beam). As the magnetization direction of the Py layer is in the thin film plane, we need to tilt the sample to $\Theta > 0^\circ$ (we selected 60°). The as-grown state consists of an equal distribution of domains with N and M parallel to the four equivalent $\langle 110 \rangle$ directions. Thus, the XMCD signal obtained for $k(x\text{-ray})$ with a component parallel $[110]$ is zero, as it averages over M parallel $[110]$ and M parallel $[-1-10]$. Once we align M (for $\Theta=0^\circ$) parallel x and then tilt the sample to $\Theta=60^\circ$, we have a component of M parallel $k(x\text{-ray})$ and measure an XMCD signal. However, if the magnetic field is aligned parallel y, it rotates M (and N) parallel y, i.e. perpendicular to $k(x\text{-ray})$. Consistently, the XMCD signal becomes zero again. The situation is different for the XMLD effect: The linear dichroism is sensitive to the component of N (or M) parallel to the electric field vector of the incoming x-rays. Furthermore, in contrast to XMCD, XMLD changes sign, if N (or M) are rotated by 90° (instead of 180° , which changes sign of the XMCD but has no effect on the XMLD).

To clarify this, we added to the METHODS section:

“The antiferromagnetic order in Mn₂Au causes an x-ray magnetic linear dichroism (XMLD), i. e. an asymmetry in the absorption of linear polarized x-rays at the Mn-L_{2,3}-edge for the polarization direction parallel and perpendicular to N [47]. **It is sensitive to the components of N, which are parallel versus perpendicular to the electric field vector of the x-ray beam, it but does not change sign upon reversal of N.**”

“The XMCD is an asymmetry in the absorption of circular polarized x-rays determined by the component of M_F parallel to the direction of the incoming x-ray beam [49]. **It does change sign upon reversal of M_F .**”

4. In the line 252, authors describe that “By SEMPA, we also verified that Mn₂Au(40 nm)/Py(4 nm)/SiNx(2 nm) samples, which were previously exposed to magnetic fields above H_c (see Fig. 2), are homogeneously magnetized”. While the experimental evidence is absent. Maybe some experimental data or more discussions are necessary.

As shown below (right panel) such images of magnetization aligned samples basically show a single grey value, in contrast to as grown samples (left panel). Most of the remaining contrast shown in the right panel is associated with sample morphology. The darker regions may be associated with domains, which were not aligned. However, as there are no domain walls etc. observable, we do not think that such an image should to be added to the manuscript.

Left: Fig. 4c from the manuscript. Right: Corresponding image of a sample previously exposed to a field larger than H_c .

5. It seems that the coercive field of Mn₂Au (100)/Py in this work is much larger than Mn₂Au (100)/Fe (Ref. 31, Adv. Mater. 24, 6374 (2012)), as well as Mn₂Au (100)/Py and Mn₂Au (100)/[Co/Pd] (Nat. Mater. 20, 800 (2021)) at room temperature. Whether the difference is from the quality of the Mn₂Au films, especially the FM/AFM interface? Please add some explanation in the revised manuscript.

Our samples are grown in (001)-orientation, i.e. with the magnetic moments in plane. The samples of Adv. Mater. 24, 6374 (2012) and Nat. Mater. 20, 800 (2021) are grown in (101)- and (103)-orientation, respectively. This may result in a different exchange coupling at the interface. Also, there is not much information on the morphology of these other Mn₂Au samples available and there is no

imaging of their AFM domain patterns published at all. Thus, we could only speculate about the origin of the small coercitive field observed by these authors. Regarding our own previous work [31], the much weaker coupling between Fe and Mn₂Au could be related to a less smooth morphology of our older samples, but also the selection of a different FM material (epitaxial Fe in [31]) can have an influence on the exchange coupling at the interface of the layers. Thus, we added your above mentioned references and the following statement to the DISCUSSION section:

“Regarding other work on Mn₂Au/FM bilayers, coercitive fields of the order of only 100 Oe or less were reported, which could be related to different growth directions of these Mn₂Au thin film [37, 38] as well as to the selection of Fe as the FM layer [31, 37], or to a different morphology.”

Then, we also present several minor comments.

1. The title “a antiferromagnetic” should be replaced by “an antiferromagnetic”, similarly, revising the spelling errors.

We corrected this.

2. In the line 238, author describe that “This enables a strong planar exchange coupling at the interface, as indicated in Fig. 4c.” Maybe the “Fig. 4c” should be replaced by “Fig.4”, that the coupling is obtained from all panels in figure 4.

We now correctly refer to Fig. 5.

3. Whether SEMPA imaging of figure 4c-e is from the as-grown state or after magnetic field? Please add description in the figure caption.

We added “as-grown” to the caption.

4. According to this manuscript, XMLD can be used to characterize ferromagnet, which is not common. What’s the difference between XMLD and XMCD for the characterization of ferromagnetic moments? Why XMCD is shown in figure 3b, while XMLD is chosen in figure 4b?

Normally, XMCD is used to investigate FMs in PEEM, as the contrast is larger than based on XMLD. As described above, XMCD is probing the component of the magnetization parallel to the incident x-ray beam. In most PEEMs, there is a shallow angle of incidence of the x-rays of about 20deg with respect to the sample surface, which is a good geometry for XMCD-PEEM of ferromagnets. However, for XMLD (as required to image AFMs) the largest contrast is obtained for perpendicular incidence of the x-ray beam. Thus we selected MAXPEEM, because, as we describe in Methods, “The SPELEEM microscope (Elmitec GmbH) of the MAXPEEM beamline provides, due to its perpendicular photon incidence on the sample surface, the highest possible AFM contrast for the in-plane orientation of the Néel vector of Mn₂Au.” The drawback of this facility is, that no in-plane ferromagnetic XMCD contrast can be obtained with perpendicular incidence (the sample cannot be tilted at MAXPEEM due to the associated image distortion). Thus, we needed to rely on the XMLD of the FM permalloy probing at the Fe-edge (Fig. 4b). However, at the spectroscopic XMCD-/XMLD-beamline at Diamond Light Source, the sample can be tilted. Thus, we could probe the FM order as shown in Fig. 3 based on the XMCD. As already discussed above, we added additional information to the METHODS section.

Reviewer #2

1) However I disagree with the main motivation point in the present version of the manuscript; e.g., in the abstract, the authors want to benefit from the well-established read-out methods of ferromagnets, while the essential advantages of antiferromagnetic spintronics are retained. In my view, the main advantage is the fast dynamics of the AFM and adding FM parts back would remove this advantage; or one has to prove experimentally that an AFM coupled to FM would retain its fast dynamics.

The fast dynamics is one of the two essential advantages of AFMs compared to FMs in spintronics. We agree that coupling the AFM to thin FM layers diminishes this advantage. By how much, however, is controllable by the strength of the exchange coupling and by the thickness of the FM layer. In our manuscript, we show that the exchange coupling is very strong, which means that the dynamics of the Py layer is expected to be relatively fast compared to ordinary FM dynamics. Furthermore, the slower dynamics of the thin FM coupled layer compared to the AFM layer alone is only affecting the speed of a memory device, if the reading process is to follow directly after the writing process. As part of a bilayer, the manipulation of the AFM with its internal THz dynamics is in principle possible, just the ultrathin Py layer is expected to follow with more or less slower dynamics, which depends of the magnitude of the exchange field. We are currently working on probing the Py dynamics (as part of Mn₂Au/Py bilayers), but this goes beyond the scope of the present manuscript. However, it is promising that first FMR experiments on a remanently magnetized Mn₂Au/Py(5 nm) bilayer show a zero field resonance at 30GHz. As the zero field resonance frequency is increasing with decreasing Py layer thickness (layer thicknesses down to 1nm are feasible) this is promising concerning fast read-out.

Anyway, we agree that these issues require a clarification in our manuscript. Thus, in the abstract we replaced: “retained” by “... while the essential advantages of antiferromagnetic spintronics are **only slightly diminished** ~~retained~~”. In the introduction, we replaced “preserved” by “the fast dynamics of the AFM as well as its resistance against disturbing magnetic fields is **only moderately diminished** ~~preserved~~.”

The other advantage of antiferromagnetic spintronics is stability in external magnetic fields, which also allows for an increased storage density (no magnetic coupling of the bits). The Mn₂Au/Py(2nm) bilayers are stable in large fields up to 0.5 T, as we demonstrate in our manuscript. Using synthetic antiferromagnetically coupled FM bilayer with zero net magnetization [26] coupled to Mn₂Au, the stability could be further increased.

Finally, the ability to readout the direction of the Néel vector (as opposed to the axis along which it is aligned) via the strongly coupled magnetization direction of the Py layer is highly beneficial for the research still required to make antiferromagnetic spintronics a real technology.

Reviewer #3

One minor suggest is to indicate that Fig.3c and Fig.3d are for Mn L edge.

We added this information to the caption of Fig. 3.

Reviewers' Comments:

Reviewer #1:

Remarks to the Author:

The authors address my previous comments and questions carefully and satisfactorily. I also found the other two reviewers evaluate the work positively. Thus I recommend its publication as it is.

Reviewer #2:

Remarks to the Author:

I believe the authors have responded adequately to the referees' concerns in the revised manuscript. I recommend the publication in the present form.